# Evaluation and Selection of Integrated Energy System Construction Scheme Equipped with Smart Energy Management and Control Platform Using Single-Valued Neutrosophic Numbers

**Junqing Wang [1], Wenhui Zhao [1,\*], Lu Qiu [2] and Puyu Yuan [3]**

[1] College of Economics and Management, Shanghai University of Electric Power, Shanghai 200090, China; junqingwang716@mail.shiep.edu.cn

[2] Jinhua Electric Power Design Institute Co., Ltd., Jinhua 321016, China; jhdlsjy@126.com

[3] College of Science, Shenyang Ligong University, Shenyang 110159, China; yuanpuyu@sylu.edu.cn

\* Correspondence: Zhao_Wenhui@shiep.edu.cn

**Abstract:** Since application of integrated energy systems (IESs) has formed a markedly increasing trend recently, selecting an appropriate integrated energy system construction scheme becomes essential to the energy supplier. This paper aims to develop a multi-criteria decision-making model for the evaluation and selection of an IES construction scheme equipped with smart energy management and control platform. Firstly, a comprehensive evaluation criteria system including economy, energy, environment, technology and service is established. The evaluation criteria system is divided into quantitative criteria denoted by interval numbers and qualitative criteria. Secondly, single-valued neutrosophic numbers are adopted to denote the qualitative criteria in the evaluation criteria system. Thirdly, in order to accommodate mixed data types consisting of both interval numbers and single-valued neutrosophic numbers, the TOPSIS (Technique for Order Preference by Similarity to an Ideal Solution) method is extended into a three-stage technique by introducing a fusion coefficient $\mu$. Then, a real case in China is evaluated through applying the proposed method. Furthermore, a comprehensive discussion is made to analyze the evaluation result and verify the reliability and stability of the method. In short, this study provides a useful tool for the energy supplier to evaluate and select a preferred IES construction scheme.

**Keywords:** integrated energy system; smart energy management and control platform; single-valued neutrosophic numbers; multi-criteria decision-making; extended TOPSIS method

## 1. Introduction

Energy is the basis for human survival. With the development of the world economy, the total energy consumption continues to increase. Nevertheless, the climate and environmental problem caused by the uncontrolled use of fossil resources deteriorates [1]. To meet the challenges of energy, environment, and climate change, developing a low-carbon economy is the only way to achieve sustainable development. As the world's largest developing country, China has been actively participating in global environment and climate change topics, increasing its national independent contribution and continuously adopting effective policies and measures to combat the problem [2,3]. At the 2015 Paris Climate Conference, the Chinese government proposed that China's carbon emissions peak by 2030 [4]. At the UN General Assembly in September 2020, President Xi Jinping further pledged to achieve carbon neutrality by 2060 [5]. To achieve this goal, more efforts are required to be made in various fields including energy supply.

### 1.1. The Development of the Integrated Energy System

The integrated energy system (IES) can provide multiple energy sources such as electricity, gas, heat and cooling at the same time to improve the consumption rate of renewable energy and the energy utilization rate of the system [6]. So it has been considered as a promising energy supply solution and has received widespread attention. Thus far, scholars from various countries have achieved great progress in the research of IES, especially in the area of IES modeling and multi-energy flow calculation [7,8], optimal scheduling [9,10], and so on. At present, the IES governs the park as an energy supply unit and integrates auxiliary energy supply systems such as distributed photovoltaic, decentralized wind power, waste heat recovery, CCHP (combined cooling, heating and power) systems, ground source heat pump, air source heat pump, sewage source heat pump, smart micro-grid, energy storage, and demand side response to achieve multi-energy complementarity [11]. Besides, attaining effective management and control of multiple energy sources is also critical for the IES. With the application of communication and information technology, smart energy management and control platform (SEMCP) is then being gradually developed [12]. Taking advantage of advanced energy Internet, big data, Internet of things, cloud service platform and artificial intelligence technology, the SEMCP can effectively break the information barriers that exist in the process of traditional energy supply and deployment, realize the intelligent management of regional energy supply and demand, and thus meet the requirements of green, low-carbon, safe, efficient, and sustainable development of the IES [13,14]. In practice, the SEMCP can not only optimize the energy scheduling strategy, but also improve the efficiency of energy management such as water, electricity and gas for energy suppliers, as well as provide better energy use services for users. Therefore, the SEMCP has already been an indispensable part of IES.

In recent years, the application of the IES equipped with the SEMCP has formed a remarkable trend in newly planned parks such as residential communities, industrial parks, school campuses, central business districts (CBDs), etc. How to evaluate and select a suitable IES construction scheme for energy supplier then becomes a critical issue, and it actually becomes a multi-criteria decision-making (MCDM) problem. Li [15] constructed an abstract mathematical model to evaluate the efficiency of an integrated energy system based on hybrid multi-attribute decision-making. Yang [16] conducted an integrated evaluation for community energy planning to select the optimal energy supply system considering the aspects of economy, environment, and society. Zhang [17] established an index evaluation model of park-level integrated energy system for a micro-grid and then selected the optimal integrated energy system composition scheme among five alternatives, considering factors of economy, reliability, energy consumption and environmental protection. As we can see, the previous research on the evaluation of the IES is limited. On the one hand, the evaluation criteria constructed in their studies pay little attention to the influence of technology innovations on IES performance, especially the latest technology applied to the SEMCP. Few practical and effective evaluation models for IES construction schemes have been constructed, either. The motivation of this paper is to establish a comprehensive evaluation criteria system and then a feasible evaluation model for IES construction schemes selection. By doing that, it cannot just provide an efficient evaluation and selection model for energy suppliers, but also promote a more extensive application of the IES and lead to further development.

### 1.2. Multi-Criteria Decision-Making Problem

MCDM problems have a wide range of applications in operation research and management science [18–20]. Due to the vagueness and uncertainty of evaluation criteria, in many cases the qualitative criteria of MCDM problems cannot be described with accurate numerical values. People generally like to directly use "excellent", "good", "poor" and other linguistic terms to describe the performance of things such as the quality of automobiles, the taste of the meal, etc. As a result, Zadeh [21] first introduced the fuzzy set to transfer uncertainty information into mathematical form. However, the fuzzy set only focuses on

the truth-membership of vague information and cannot deal with the falsity-membership and indeterminacy-membership. Thus, the intuitionistic fuzzy set (IFS) was proposed by Atanassov [22] with both truth-membership and falsity-membership in 1986. Nevertheless, it still could not accurately deal with the uncertainty of information in various practical problems. Therefore, Smarandache [23] proposed the concepts of neutrosophic logic and neutrosophic set from a philosophical point of view. In order to apply neutrosophic set to practical engineering and scientific applications, Wang [24] introduced single-value neutrosophic set (SVNS), and Ye developed simplified neutrosophic set [25] and interval neutrosophic set [26]. In the neutrosophic set, the uncertainty can be explicitly quantified through membership functions. The truth-membership, the indeterminacy-membership, and the falsity-membership are all considered and independent of each other in SVNS. In addition, many scholars combine neutrosophic set with other ranking methods, such as ELECTRE [27] or Technique for Order Preference by Similarity to an Ideal Solution (TOPSIS) [28] to solve the MCDM problem. In practice, SVNS can not only be used to denote uncertain information during evaluation, but can also be used to determine the weight of evaluation criteria through the neutrosophic entropy method [29]. Furthermore, SVNS can be fuzzied into a crisp number, if necessary [28]. Actually, due to its flexibility and practicality, SVNS has been applied to many fields. Zavadskas [30] conducted an assessment of alternative sites for the construction of a waste incineration plant with single-valued neutrosophic set. Long [31] proposed a novel restoration methods selection approach for wood components of Chinese ancient architectures with single-valued neutrosophic sets. It can be seen that SVNS has good practicality in real situations for solving MCDM problems. However, to the best of my knowledge, SVNS has not been applied to evaluate or select any kind of energy construction scheme yet.

### 1.3. Problem of Previous Study and Contribution of this Paper

Through the introduced background and literature review above, it can be noticed that some critical problems have not been well solved by previous studies: (1) With regard to evaluation criteria system of the IES, previous related studies lack considerations of the rapid technology development. No updated criteria system has yet been constructed for evaluating and selecting an IES construction scheme. Especially, the evaluation criteria of the SEMCP has never before been considered to be added to the criteria system. (2) When the evaluation criteria system of the MCDM problem consists of qualitative criteria, the fuzzy numbers must be introduced to denote them. Then, how to reduce the information loss of qualitative criteria as little as possible is always a difficult issue. (3) When the decision matrix of the MCDM problem is composed of different data types, most previous study choose to transfer different data types into the same data type. However, it inevitably causes lots of information loss when the transformation procedure progresses.

Thus, in order to solve the above problems, the main contribution of this paper can be summarized as follows: (1) Through in-depth study of the IES and considering the impact of relevant technology developments on the IES, the technology criteria and service criteria of the SEMCP are added into the criteria system for evaluating the IES construction scheme. This paper establishes a comprehensive evaluation criteria system that consists of not only traditional quantitative criteria, including economy, energy and environment, but also innovative qualitative criteria including technology criteria and service criteria. (2) The single-valued neutrosophic set is introduced into the evaluation of an IES construction scheme. In order to maintain the information completeness of the decision makers' linguistic assessment of qualitative criteria, we adopt single-valued neutrosophic numbers (SVNNs) to quantify the linguistic variables given by decision makers. Obviously, truth-membership degree, indeterminacy-membership degree, and falsity-membership degree in SVNNs can fully denote the decision maker's individual judgment and are closer to the real decision-making environment. (3) The TOPSIS method is the most widely used to solve MCDM problems, which determines a solution with the shortest distance from the ideal solution and the farthest distance from the negative

ideal solution. Since the quantitative criteria are described by interval numbers and the qualitative criteria are described by SVNNs in this paper, the classical TOPSIS method is extended to apply to an evaluation criteria system with mixed data types. (4) By applying the extended TOPSIS method proposed in this paper, an empirical study is conducted. In addition, a detailed discussion including sensitivity analysis, scenario analysis and comparative analysis is provided to analyze the evaluation result.

The rest of this paper is organized as follows: A comprehensive evaluation criteria system for the IES is established and each sub-criterion is described in detail in Section 2. The extended TOPSIS method for SVNNs and interval numbers is elaborated in Section 3. Section 4 gives an empirical study in China, and the evaluation result is discussed in Section 5. The last section concludes this paper.

## 2. Materials

With the rapid development of various technologies of the IES, the core role of the SEMCP has become more prominent, and the criteria system for evaluating the IES construction scheme has also been optimized. The traditional evaluation criteria system for the IES only considers quantitative criteria such as economic benefits, energy benefits, and environmental benefits, and lacks consideration of the innovative qualitative criteria for evaluating the embedded SEMCP, such as technological advancement and user service level. Consequently, a comprehensive evaluation criteria system for the IES covering economy, energy, environment, technology and service criteria was established, as shown in Table 1. Each evaluation sub-criterion is elaborated as follows.

**Table 1.** The established evaluation criteria system of the integrated energy system (IES).

| Criteria | Sub-Criteria (Unit) | |
|---|---|---|
| (a) economy | (C1) Construction cost (RMB 10,000) | quantitative |
| | (C2) Operations and maintenance cost (RMB 10,000) | quantitative |
| (b) energy | (C3) Primary energy conservation (ton of standard coal equivalent) | quantitative |
| | (C4) Renewable energy utilization (%) | quantitative |
| (c) environment | (C5) $CO_2$ emission reduction (ton) | quantitative |
| | (C6) Emission reduction of other pollutants (ton) | quantitative |
| (d) technology | (C7) Comprehensive monitoring capability | qualitative |
| | (C8) Energy regulating and stabilizing capability | qualitative |
| | (C9) Analysis and decision-making capability | qualitative |
| | (C10) Intelligent operation and maintenance capability | qualitative |
| (e) service | (C11) Informatization level of service | qualitative |
| | (C12) Satisfaction degree of user service | qualitative |

### 2.1. Economy Criteria

- Construction cost (C1): The construction cost of the IES covers not only the expenditure of photovoltaic modules, energy storage equipment, CCHP system, natural gas pipeline network, ground source heat pump and other energy equipment needed, but also the expenditure of developing the SEMCP embedded in the IES. The SEMCP includes various meters, sensors, terminal equipment, application servers, database servers, collection servers, and operating systems on the servers and application software on terminals.
- Operations and maintenance (O&M) cost (C2): The annual O&M cost consists of the laborers' salaries and the maintenance expenditure of equipment and software in the IES.

### 2.2. Energy Criteria

- Primary energy conservation (C3): Primary energy conservation refers to the amount of primary energy saved through the integrated energy management and multiple

energy sources optimal dispatch of the IES, compared to the traditional energy supply system in one year. The unit of this indicator is converted into standard coal.

- Renewable energy utilization (C4): Renewable energy utilization refers to the proportion of renewable energy consumption in total energy consumption. The renewable energy such as solar resources, wind resources and ground heat resources is considered to be exploited in the IES depending on the local resource circumstances. In cooperation with the energy storage system and multi-energy collaborative optimization model of the IES, the consumption of distributed renewable energy can be increased in the park. Renewable energy utilization is an indispensable evaluation indicator for evaluating an energy scheme.

## 2.3. Environment Criteria

- Carbon dioxide emission reduction (C5): The IES can reduce carbon dioxide emissions by raising the utilization of renewable energy and improving energy conversion efficiency, thus reducing the consumption of fossil energy, which results in most greenhouse gas emissions. Saving 1 kg of standard coal is equivalent to reducing carbon dioxide emissions by 2.493 kg.
- Emission reduction of other pollutants (C6): The application of the IES can also reduce emission of other pollutants, including $SO_2$, $NO_x$ gas, etc. The environmental pollution caused by these harmful gas emissions cannot be ignored, either. Saving 1 kg of standard coal is equivalent to reducing 0.038 kg of $SO_2$ and 0.075 kg of $NO_x$.

## 2.4. Technology Criteria

- Technology criteria are indicators for evaluating the SEMCP. By consulting a large number of pertinent literatures and professional experts' advice, four basic capabilities were chosen to evaluate the technological advancement of the SEMCP, including real-time monitoring capability, multi-energy optimal dispatch capability, energy data analysis capability, and intelligent operation and maintenance capability [32–36].
- Comprehensive monitoring capability (C7): Comprehensive monitoring capability is mainly reflected in monitoring the real-time operating status of critical equipment. The power distribution monitoring module can deeply sense the operating status of the source-grid-load-storage and guarantee a stable and reliable power supply. The energy consumption monitoring module can monitor the energy consumption information of gas, electricity, heat and cooling in buildings, and support the energy operation and management of the energy supply center.
- Energy regulating and stabilizing capability (C8): Energy regulation includes two parts: intelligent dispatch and load control. It can optimize the control strategy and improve the accuracy of the control strategy through real-time sensing of load changes and energy analysis. The unit commitment of generators, storage and load demand within the IES is also critical to create a cost-effective, reliable and environmentally friendly energy provision system [37,38]. The pressure on the large power grid during load peak can also be relieved by regulating interruptible loads. The platform adopts a new set of strategies modifying traditional generation control algorithms to raise the reliability of the IES [39].
- Analysis and decision-making capability (C9): Energy analysis includes energy utilization level analysis, peak and valley electricity analysis, building energy consumption analysis, energy consumption comparison analysis and social benefit analysis. By copying the multi-energy metering data of gas, electricity, heat and cooling in the park, regular energy operation reports are generated to help energy suppliers understand the overall energy operating condition of the entire park.
- Intelligent operation and maintenance capability (C10): Intelligent operation and maintenance includes three parts: asset ledger module, operation and maintenance information module and operation and maintenance analysis module. The asset ledger module can perceive important asset information including equipment number,

equipment model and manufacturer throughout its life cycle. The operation and maintenance information module can formulate inspection plans, dispatch inspection tasks, record fault information and perform online management of fault information. The operation and maintenance analysis module can realize automatic fault identification, fault cause analysis, fault impact analysis and automatically give fault handling suggestions.

### 2.5. Service Criteria

- Informatization level of service (C11): Users can query daily household energy use data on the mobile APP, pay the energy bills online, check the status and location of charging piles in the park, and can also receive some energy use suggestions. The energy supplier can view the total energy load of the entire park and the operating status of important equipment at any time through the display screen in the energy supply center.
- Satisfaction degree of user service (C12): With the continuous development of the energy market, users' energy demand has gradually shown differentiated and diversified characteristics, embodied in two dimensions: basic energy demand and value-added energy demand. The basic energy demand is the consumer's demand for electricity, gas, heat, cooling and other energy consumption. Value-added energy demand is the user's incremental demand for improving energy use experience and energy use benefits, such as saving energy costs, improving energy use efficiency and consuming renewable energy [40,41].

## 3. Methods

### 3.1. Preliminary

3.1.1. Single-Valued Neutrosophic Set

Some basic definitions, operations, and properties regarding single-valued neutrosophic sets are provided in this section.

**Definition 1.** *Let X be a universal space of points (objects), with a generic element of X denoted by x. A single-valued neutrosophic set (SVNS)* [42] *is characterized by a truth-membership function* $T_A(x)$ *, an indeterminacy-membership function* $I_A(x)$, *and a falsity-membership function* $F_A(x)$ *with* $T_A(x), I_A(x), F_A(x) \in [0,1]$ *, for all* $x \in X$ *. SVNS can be represented with the notation* $A = \{\langle x, T_A(x), I_A(x), F_A(x)\rangle | x \in X\}$.

**Definition 2.** *Let* $A = \langle T_A(x), I_A(x), F_A(x)\rangle$ *and* $B = \langle T_B(x), I_B(x), F_B(x)\rangle$ *be two SVNSs, for all* $x \in X$ *and then the operations can be defined as follows* [43]:
*(a)* $A \subseteq B \Leftrightarrow T_A(x) \leq T_B(x), I_A(x) \geq I_B(x), F_A(x) \geq F_B(x)$ *,*
*(b)* $A = B \Leftrightarrow A \subseteq B$ *and* $B \subseteq A$ *,*
*(c)* $A^c = \langle F_A(x), 1 - I_A(x), T_A(x)\rangle$ *, where* $A^c$ *denotes the complement of* $A$,
*(d)* $A \cup B = \langle \max(T_A(x), T_B(x)), \min(I_A(x), I_B(x)), \min(F_A(x), F_B(x))\rangle$,
*(e)* $A \cap B = \langle \min(T_A(x), T_B(x)), \max(I_A(x), I_B(x)), \max(F_A(x), F_B(x))\rangle$.

**Definition 3.** *Let A and B be two SVNSs, for all* $x \in X$, $\forall \lambda \in R$ *and* $\lambda > 0$ [43], *then*
*(a)* $\lambda A = \langle 1 - (1 - T_A(x))^\lambda, I_A(x)^\lambda, F_A(x)^\lambda\rangle$,
*(b)* $A \oplus B = \langle T_A(x) + T_B(x) - T_A(x) \cdot T_B(x), I_A(x) \cdot I_B(x), F_A(x) \cdot F_B(x)\rangle$,
*(c)* $A \otimes B = \left\langle \begin{array}{c} T_A(x) \cdot T_B(x), I_A(x) + I_B(x) - I_A(x) \cdot I_B(x), F_A(x) + F_B(x) - \\ F_A(x) \cdot F_B(x) \end{array} \right\rangle$.

**Definition 4.** *Fuzzification of SVNS* $A = \{\langle x, T_A(x), I_A(x), F_A(x)\rangle | x \in X\}$ *is defined as a process of mapping A into fuzzy set* $F = \{x | \mu_F(x) | x \in X\}$ *, and the equivalent fuzzy membership degree is as* [28]:

$$\mu_F(x) = 1 - \sqrt{\left\{(1 - T_A(x))^2 + (I_A(x))^2 + (F_A(x))^2\right\}/3}, x \in X. \tag{1}$$

**Definition 5.** *Let* $A = \{(x_1 | \langle T_A(x_1), I_A(x_1), F_A(x_1)\rangle), \dots, (x_n | \langle T_A(x_n), I_A(x_n), F_A(x_n)\rangle)\}$ *and* $B = \{(x_1 | \langle T_B(x_1), I_B(x_1), F_B(x_1)\rangle), \dots, (x_n | \langle T_B(x_n), I_B(x_n), F_B(x_n)\rangle)\}$ *be two SVNSs for* $x_i \in X(i = 1, 2, \dots, n)$. *Suppose the weight* $\omega_i$ *of the element* $x_i$, *with* $\omega_i \geq 0$ *and* $\sum_{i=1}^{n} \omega_i = 1$. *Then the weighted Euclidean distance between two SVNSs A and B can be defined as follows* [44]:

$$D_E(A, B) = \left\{\sum_{i=1}^{n} \omega_i [(T_A(x_i) - T_B(x_i))^2 + (I_A(x_i) - I_B(x_i))^2 + (F_A(x_i) - F_B(x_i))^2]\right\}^{\frac{1}{2}}. \tag{2}$$

The normalized weighted Euclidean distance between two SVNSs A and B can be defined as follows [45,46]:

$$D_E^N(A, B) = \left\{\frac{1}{3n}\sum_{i=1}^{n} \omega_i [(T_A(x_i) - T_B(x_i))^2 + (I_A(x_i) - I_B(x_i))^2 + (F_A(x_i) - F_B(x_i))^2]\right\}^{\frac{1}{2}}. \tag{3}$$

3.1.2. Neutrosophic Entropy

**Definition 6.** *According to Majumder* [29], *for single-valued neutrosophic set* $A = \langle T_A(x_i), I_A(x_i), F_A(x_i)\rangle$, *the entropy on neutrosophic set A is computed as follows*:

$$E(A) = 1 - \frac{1}{n}\sum_{i=1}^{n} \{(T_A(x_i) + F_A(x_i)) | I_A(x_i) - I_{A^c}(x_i)|\}. \tag{4}$$

Entropy represents the uncertainty of the criterion value. In the process of decision-making, if the value of the criterion can be represented by a single-valued neutrosophic number, the criterion weight can be calculated by Equation (5) [47]:

$$\omega_j = \frac{1 - E(A_j)}{\sum\limits_{j=1}^{n}(1 - E(A_j))}, j = 1, 2, \cdots, n. \tag{5}$$

3.1.3. Interval Numbers

**Definition 7.** *Let z be a non-negative interval number* [48], *which has the following form:* $z = [a^L, a^U] = \{x | 0 \leq a^L \leq x \leq a^U, a^L, a^U \in R\}$. *If* $a^L = a^U$ *, z is degenerated into a non-negative real number. For any two interval numbers* $\widetilde{a} = [a^L, a^U]$ *and* $\widetilde{b} = [b^L, b^U]$ *, then the operations can be defined as follows:*
  *(a)* $\widetilde{a} + \widetilde{b} = [a^L + b^L, a^U + b^U]$ *,*
  *(b)* $\widetilde{a} = \widetilde{b} \Leftrightarrow a^L = b^L$ *and* $a^U = b^U$ *,*
  *(c)* $k\widetilde{a} = [ka^L, ka^U], k \geq 0$ *.*

**Definition 8.** *Let* $\widetilde{A} = \{[a_1^L, a_1^U], [a_2^L, a_2^U], \cdots, [a_n^L, a_n^U]\}$ *and* $\widetilde{B} = \{[b_1^L, b_1^U], [b_2^L, b_2^U], \cdots, [b_n^L, b_n^U]\}$ *be two interval number sets, and the weight corresponding to each element in the set is* $\omega_i$, *i = 1, 2, $\cdots$, n. Then, the weighted normalized Euclidean distance between* $\widetilde{A}$ *and* $\widetilde{B}$ *can be defined as follows* [48]:

$$D_E^N(\widetilde{A}, \widetilde{B}) = \sqrt{\frac{1}{2n}\sum_{i=1}^{n} \omega_i [(a_i^U - b_i^U)^2 + (a_i^L - b_i^L)^2]}. \tag{6}$$

### 3.2. Extended TOPSIS Method for SVNSs and Interval Numbers

The evaluation method proposed in this paper extends the classical TOPSIS method. As we know, the classical TOPSIS method can only be used to evaluate the criteria system whose criterion value is expressed by the same data type, such as interval number or SVNNs. But in many cases, the evaluation criteria system consists of both qualitative criteria and quantitative criteria, which are usually expressed by different kinds of data types. Thus, we adopted a three-stage technique to make the TOPSIS method applicable to a criteria system made up of mixed data types. Firstly, we applied the classical TOPSIS method to quantitative criteria denoted by interval number and obtained the evaluation result of this stage, see Section 3.2.1. Secondly, we applied the classical TOPSIS method to qualitative criteria denoted by SVNNs and obtained the phased evaluation result similarly, see Section 3.2.2. As a result, we integrated the results obtained in the previous two stages to acquire the final evaluation result, see Section 3.2.3.

Consider an MCDM problem with $t$ decision makers, $m$ alternatives and $n$ evaluation criteria. Let $A = \{A_1, A_2, \cdots, A_m\}$ be a discrete set of alternatives, $C = \{C_1, C_2, \cdots, C_n\}$ be the set of criteria, and $D = \{D_1, D_2, \cdots, D_t\}$ be the set of decision makers whose corresponding decision weight set is $e = \{e_1, e_2, \cdots, e_t\}$, where $\sum_{k=1}^{t} e_k = 1, 0 \le e_k \le 1$. The former $s$ criteria of $n$ evaluation criteria are quantitative criteria, which are denoted by interval numbers as $[a^L, a^U]$, and the latter $(n\text{-}s)$ criteria are qualitative ones, which are denoted by SVNNs as $\langle T, I, F \rangle$. The decision matrix associated with the alternatives with respect to each criterion for the MCDM problems can be presented in following form.

$$\mathbf{D} = \langle d_{ij} \rangle_{m \times n} =$$

$$
\begin{array}{c}
\begin{array}{cccccccc}
\quad C_1 & \quad C_2 & \cdots & C_s & \quad C_{s+1} & \quad C_{s+2} & \cdots & \quad C_n
\end{array} \\
\begin{array}{c} A_1 \\ A_2 \\ \vdots \\ A_m \end{array}
\left(
\begin{array}{cccccccc}
[a_{11}^L, a_{11}^U] & [a_{12}^L, a_{12}^U] & \cdots & [a_{1s}^L, a_{1s}^U] & \langle T_{1s+1}, I_{1s+1}, F_{1s+1} \rangle & \langle T_{1s+2}, I_{1s+2}, F_{1s+2} \rangle & \cdots & \langle T_{1n}, I_{1n}, F_{1n} \rangle \\
[a_{21}^L, a_{21}^U] & [a_{22}^L, a_{22}^U] & \cdots & [a_{2s}^L, a_{2s}^U] & \langle T_{2s+1}, I_{2s+1}, F_{2s+1} \rangle & \langle T_{2s+2}, I_{2s+2}, F_{2s+2} \rangle & \cdots & \langle T_{2n}, I_{2n}, F_{2n} \rangle \\
\vdots & \vdots & \vdots & \vdots & \vdots & \vdots & \ddots & \vdots \\
[a_{m1}^L, a_{m1}^U] & [a_{m2}^L, a_{m2}^U] & \cdots & [a_{ms}^L, a_{ms}^U] & \langle T_{ms+1}, I_{ms+1}, F_{ms+1} \rangle & \langle T_{ms+2}, I_{ms+2}, F_{ms+2} \rangle & \cdots & \langle T_{mn}, I_{mn}, F_{mn} \rangle
\end{array}
\right).
\end{array}
$$

#### 3.2.1. Calculation and Analysis for Quantitative Criteria

Step 1: Construct the decision matrix of the quantitative criteria.

Decision makers construct the decision matrix of the quantitative criteria by collecting data of the alternatives with respect to corresponding evaluation criteria. The decision matrix of the quantitative criteria is in the form as follows.

$$
D_1 = \langle d_{ij} \rangle_{m \times s} =
\begin{array}{c}
\begin{array}{c} A_1 \\ A_2 \\ \vdots \\ A_m \end{array}
\begin{array}{cccc}
C_1 & C_2 & \cdots & C_s
\end{array} \\
\left(
\begin{array}{cccc}
d_{11} & d_{12} & \cdots & d_{1s} \\
d_{21} & d_{22} & \cdots & d_{2s} \\
\vdots & \vdots & \ddots & \vdots \\
d_{m1} & d_{m2} & \cdots & d_{ms}
\end{array}
\right)
\end{array}
=
\begin{array}{c}
\begin{array}{c} A_1 \\ A_2 \\ \vdots \\ A_m \end{array}
\begin{array}{cccc}
\quad C_1 & \quad C_2 & \cdots & \quad C_s
\end{array} \\
\left(
\begin{array}{cccc}
[a_{11}^L, a_{11}^U] & [a_{12}^L, a_{12}^U] & \cdots & [a_{1s}^L, a_{1s}^U] \\
[a_{21}^L, a_{21}^U] & [a_{22}^L, a_{22}^U] & \cdots & [a_{2s}^L, a_{2s}^U] \\
\vdots & \vdots & \ddots & \vdots \\
[a_{m1}^L, a_{m1}^U] & [a_{m2}^L, a_{m2}^U] & \cdots & [a_{ms}^L, a_{ms}^U]
\end{array}
\right)
\end{array}
$$

Step 2: Normalize the decision matrix of the quantitative criteria.

The data in the decision matrix are normalized to unify different measurement scales. The normalization equation for interval numbers is shown below. Equation (7) is for cost criteria and Equation (8) is for benefit criteria.

$$
\begin{cases}
r_{ij}^L = (1/a_{ij}^U) / \sqrt{\sum_{i}^{n} (1/a_{ij}^L)^2} \\
r_{ij}^U = (1/a_{ij}^L) / \sqrt{\sum_{i}^{n} (1/a_{ij}^U)^2}
\end{cases}
(1 \le i \le m, 1 \le j \le s), \text{ for cost criteria,} \tag{7}
$$

$$\begin{cases} r_{ij}^L = a_{ij}^L / \sqrt{\sum\limits_{i}^{n} (a_{ij}^U)^2} \\ r_{ij}^U = a_{ij}^U / \sqrt{\sum\limits_{i}^{n} (a_{ij}^L)^2} \end{cases} (1 \le i \le m, 1 \le j \le s), \text{ for benefit criteria.} \quad (8)$$

Then, we obtain the normalization matrix $D_1^N = \left\langle d_{ij}^N \right\rangle_{m \times s}$ as follows.

$$D_1^N = \left\langle d_{ij}^N \right\rangle_{m \times s} = \begin{array}{c} \\ A_1 \\ A_2 \\ \vdots \\ A_m \end{array} \begin{array}{cccc} C_1 & C_2 & \cdots & C_s \\ \begin{pmatrix} [r_{11}^L, r_{11}^U] & [r_{12}^L, r_{12}^U] & \cdots & [r_{1s}^L, r_{1s}^U] \\ [r_{21}^L, a_{21}^U] & [r_{22}^L, r_{22}^U] & \cdots & [r_{2s}^L, r_{2s}^U] \\ \vdots & \vdots & \ddots & \vdots \\ [r_{m1}^L, r_{m1}^U] & [r_{m2}^L, r_{m2}^U] & \cdots & [r_{ms}^L, r_{ms}^U] \end{pmatrix} \end{array}$$

Step 3: Obtain the relative positive ideal solution $y^{I+}$ and the relative negative ideal solution $y^{I-}$ of the quantitative criteria.

$$\begin{cases} y^{I+} = (y_1^+, y_2^+, \cdots, y_s^+) \\ y^{I-} = (y_1^-, y_2^-, \cdots, y_s^-) \end{cases}, \quad (9)$$

$$\begin{cases} y_j^+ = [y_j^{+L}, y_j^{+U}] = [\max\limits_{1 \le i \le m}(r_{ij}^L), \max\limits_{1 \le i \le m}(r_{ij}^U)], (1 \le j \le s) \\ y_j^- = [y_j^{-L}, y_j^{-U}] = [\min\limits_{1 \le i \le m}(r_{ij}^L), \min\limits_{1 \le i \le m}(r_{ij}^U)], (1 \le j \le s) \end{cases}. \quad (10)$$

Step 4: Determine the weights of quantitative criteria.

Decision maker have their own views on the importance of different evaluation criteria. A linguistic term set along with SVNNs is offered in Table 2. Firstly, decision makers can rate the importance of each quantitative criterion with the linguistic terms. The weight evaluation matrix of quantitative criteria can be obtained. Then, by using Equation (4) introduced in Section 3.1.2, we can compute the entropy value of each criterion based on the weight evaluation matrix. Finally, by using Equation (5) we can acquire the weight $\omega_j$ of each quantitative criterion.

**Table 2.** Linguistic terms with single-valued neutrosophic numbers (SVNNs) for rating the importance of evaluation criteria and decision makers.

| Linguistic Terms | SVNNs |
|---|---|
| Very important (VI) | <0.90,0.10,0.10> |
| Important (I) | <0.80,0.20,0.15> |
| Medium (M) | <0.50,0.40,0.45> |
| Unimportant (UI) | <0.35,0.60,0.70> |
| Very unimportant (VUI) | <0.10,0.80,0.90> |

Step 5: Determine the distance measure of each alternative from the relative positive ideal solution (RPIS) and the relative negative ideal solution (RNIS) for quantitative criteria.

According to Equation (6), the weighted normalized Euclidean distance measure of each alternative for quantitative criteria from the RPIS for quantitative criteria can be computed as follows:

$$d_i^{I+} = \sqrt{\frac{1}{2s} \sum_{j=1}^{s} \omega_j [(y_j^{+U} - r_{ij}^U)^2 + (y_j^{+L} - r_{ij}^L)^2]} i = 1, 2, \cdots, m. \quad (11)$$

Similarly, the weighted normalized Euclidean distance measure of each alternative for quantitative criteria from the RNIS for quantitative criteria can be computed as:

$$d_i^{I-} = \sqrt{\frac{1}{2s}\sum_{j=1}^{s}\omega_j[(y_j^{-U} - r_{ij}^U)^2 + (y_j^{-L} - r_{ij}^L)^2]} \, i = 1, 2, \cdots, m. \tag{12}$$

Step 6: Calculate the relative closeness coefficient to the ideal solution of quantitative criteria.

$$c_i^I = \frac{d_i^{I-}}{d_i^{I+} + d_i^{I-}}, i = 1, 2, \cdots, m. \tag{13}$$

### 3.2.2. Calculation and Analysis for Qualitative Criteria

Step 1: Determine the decision matrix of the qualitative criteria.

Decision makers are more inclined to use fuzzy numbers such as linguistic terms with multiple granularities to assess qualitative criteria. Decision makers can use the linguistic terms along with the SVNNs defined in Table 3 to rate alternatives with respect to each qualitative criterion. Then, the decision matrix of the qualitative criteria is obtained.

**Table 3.** Linguistic terms with SVNNs for rating the alternatives with respect to qualitative criteria.

| Linguistic Terms | SVNNs |
|---|---|
| Extremely good(EG) | <1.00,0.00,0.00> |
| Very good(VG) | <0.90,0.10,0.05> |
| Good(G) | <0.80,0.20,0.15> |
| Medium good(MG) | <0.65,0.35,0.30> |
| Medium(M) | <0.50,0.50,0.45> |
| Medium bad(MB) | <0.35,0.65,0.60> |
| Bad(B) | <0.20,0.65,0.80> |
| Very bad(VB) | <0.10,0.85,0.90> |
| Extremely bad(EB) | <0.05,0.90,0.95> |

Step 2: Determine the weight of decision makers.

Due to different experiences and backgrounds, the importance of the decision makers in a committee differs. Each decision maker is given an importance assessment according to the linguistic terms in Table 2. Then, by applying Equation (1), the weight of *t* decision makers can be calculated as:

$$e_k = \frac{u_k}{\sum\limits_{k=1}^{t} u_k} = \frac{1 - \sqrt{\left\{(1-T_k)^2 + (I_k) + (F_k)\right\}/3}}{\sum\limits_{k=1}^{t}\left\{1 - \sqrt{\left\{(1-T_k)^2 + (I_k) + (F_k)\right\}/3}\right\}},$$

$$\sum\limits_{k=1}^{t} e_k = 1, 0 \le e_k \le 1, 1 \le k \le t. \tag{14}$$

Step 3: Obtain the aggregated neutrosophic decision matrix of qualitative criteria.

Based on the decision makers' weight obtained in Step 2, the decision matrix that aggregates decision makers' weight can be calculated by using the single-valued neutrosophic weighted averaging (SVNWA) aggregation operator proposed by Ye [25] as follows:

$$d_{ij} = e_1 d_{ij}^{(1)} \oplus e_2 d_{ij}^{(2)} \oplus \cdots e_t d_{ij}^{(t)} = \left\langle 1 - \prod_{k=1}^{t}(1 - T_{ij}^k)^{e_k}, \prod_{k=1}^{t}(I_{ij}^k)^{e_k}, \prod_{k=1}^{t}(F_{ij}^k)^{e_k}\right\rangle.$$

$$i = 1, 2, \cdots, m; j = s+1, s+2, \cdots, n. \tag{15}$$

Then, the aggregated neutrosophic decision matrix of qualitative criteria is displayed as follows:

$$D_2 = \left\langle d_{ij} \right\rangle_{m \times (n-s)} = \left\langle T_{ij}, I_{ij}, F_{ij} \right\rangle =$$

$$
\begin{array}{ccccc}
 & C_{s+1} & C_{s+2} & \cdots & C_n \\
\begin{array}{c} A_1 \\ A_2 \\ \vdots \\ A_m \end{array} &
\left(
\begin{array}{c}
\left\langle T_{1s+1}, I_{1s+1}, F_{1s+1} \right\rangle \\
\left\langle T_{2s+1}, I_{2s+1}, F_{2s+1} \right\rangle \\
\vdots \\
\left\langle T_{ms+1}, I_{ms+1}, F_{ms+1} \right\rangle
\end{array}
\right. &
\begin{array}{c}
\left\langle T_{1s+2}, I_{1s+2}, F_{1s+2} \right\rangle \\
\left\langle T_{2s+2}, I_{2s+2}, F_{2s+2} \right\rangle \\
\vdots \\
\left\langle T_{ms+2}, I_{ms+2}, F_{ms+2} \right\rangle
\end{array} &
\begin{array}{c}
\cdots \\
\cdots \\
\ddots \\
\cdots
\end{array} &
\left.
\begin{array}{c}
\left\langle T_{1n}, I_{1n}, F_{1n} \right\rangle \\
\left\langle T_{2n}, I_{2n}, F_{2n} \right\rangle \\
\vdots \\
\left\langle T_{mn}, I_{mn}, F_{mn} \right\rangle
\end{array}
\right)
\end{array}.
$$

Step 4: Obtain the relative positive ideal solution $y^{N+}$ and the relative negative ideal solution $y^{N-}$ of qualitative criteria.

$$
\begin{cases}
y^{N+} = (y^+_{s+1}, y^+_{s+2}, \cdots, y^+_n) \\
y^{N-} = (y^-_{s+1}, y^-_{s+2}, \cdots, y^-_n)
\end{cases},
\tag{16}
$$

$$
\begin{cases}
y^{N+}_j = \left\langle T^+_j, I^+_j, F^+_j \right\rangle = \left\langle \max_i T_{ij}, \min_i I_{ij}, \min_i F_{ij} \right\rangle, i = 1, 2, \cdots, m \,; j = s+1, s+2, \cdots, n \\
y^{N-}_j = \left\langle T^-_j, I^-_j, F^-_j \right\rangle = \left\langle \min_i T_{ij}, \max_i I_{ij}, \max_i F_{ij} \right\rangle, i = 1, 2, \cdots, m \,; j = s+1, s+2, \cdots, n
\end{cases}
\tag{17}
$$

Step 5: Determine the weight of qualitative criteria.

The procedure for determining the weight of qualitative criteria is the same as that of quantitative criteria in Section 3.2.1, Step 4.

Step 6: Calculate the distance measure of each alternative from the relative positive ideal solution (RPIS) and the relative negative ideal solution (RNIS) for qualitative criteria.

According to Equation (3), the weighted normalized Euclidean distance measure of each alternative from the RPIS for qualitative criteria can be expressed as follows:

$$
d^{N+}_i = \sqrt{\frac{1}{3(n-s)} \sum_{j=s+1}^{n} \omega_j \left[ (T^+_j - T_{ij})^2 + (I^+_j - I_{ij})^2 + (F^+_j - F_{ij})^2 \right]}, i = 1, 2, \cdots, m.
\tag{18}
$$

Similarly, the weighted normalized Euclidean distance measure of each alternative from the RNIS for qualitative criteria can be expressed as:

$$
d^{N-}_i = \sqrt{\frac{1}{3(n-s)} \sum_{j=s+1}^{n} \omega_j \left[ (T^-_j - T_{ij})^2 + (I^-_j - I_{ij})^2 + (F^-_j - F_{ij})^2 \right]}, i = 1, 2, \cdots, m.
\tag{19}
$$

Step 7: Calculate the relative closeness coefficient to the ideal solution of qualitative criteria.

$$
c^N_i = \frac{d^{N-}_i}{d^{N+}_i + d^{N-}_i}, i = 1, 2, \cdots, m.
\tag{20}
$$

3.2.3. Acquisition of the Comprehensive Relative Closeness Coefficient and Ranking the Alternatives

As we have already acquired the relative closeness coefficient to the ideal solution of quantitative criteria $c^I_i$ in Section 3.2.1 and the relative closeness coefficient to the ideal solution of qualitative criteria $c^N_i$ in Section 3.2.2, the comprehensive relative closeness coefficient $c_i$ can be obtained by fusing them with a coefficient $\mu$ as Equation (21), where $1 > \mu > 0$. Finally, the alternatives are ranked by the comprehensive relative closeness coefficient $c_i$. The larger the value of $c_i$, the better the reflection of alternative $A_i$.

$$
c_i = \mu c^I_i + (1 - \mu) c^N_i, i = 1, 2, \cdots, m.
\tag{21}
$$

## 4. A Case Study

### 4.1. Problem Statement

Located in a newly planned residential community in Shanghai Pudong New Area, China, an energy supplier plans to build an IES equipped with an SEMCP for the residential park. The area of the park covers about 460 mμ, and the construction area is 38,600 square meters. According to the data from Solargis website, the direct normal irradiation (DNI) of solar light in this area is 1235 kWh/m$^2$, which meets the requirement for developing photovoltaic (PV) power generation. The region also reaches the shallow geothermal energy development and utilization standard, which means geothermal heat energy can be utilized for this park. The natural gas can be supplied sufficiently to the park as well.

According to local resources and geographical characteristics, the IES construction scheme of the newly residential park could consider a hybrid combination of energy supply strategy such as a hybrid energy storage system, the power grid, solar photovoltaic (PV), combined cooling, heating and power (CCHP) system, and ground source heat pump system (GSHPs). The energy supplier planned to select an appropriate IES construction scheme for the park through public bidding. After the invitation for bidding documents was issued, a total of four bidding proposals provided by four different companies were received. A brief description of these schemes can be seen in Table 4. The construction scheme of the first company A1 chose a combination of GSHP system and solar photovoltaic as the energy supply source. The alternative scheme A2 employed a CCHP system and solar photovoltaic. The construction schemes provided by the other two companies A3 and A4 both took the strategy of a combination of CCHP system, solar photovoltaic and GSHP system. Besides, all four alternative schemes made the power grid as the main energy supply and took the hybrid energy storage system as a tool of energy regulation. In addition, each alternative construction scheme was equipped with a unique SEMCP developed by the company that applied to its IES construction scheme most appropriately, such as the SEMCP-A1 for alternative A1, SEMCP-A2 for alternative A2, and so on.

**Table 4.** Four alternative IES construction schemes for a residential park.

| Alternative | Description of Each IES Construction Scheme |
|---|---|
| A1 | PV-GSHP system: Hybrid energy storage system (HESS) + GSHP system + PV unit + SEMCP-A1 |
| A2 | PV-gas system: HESS + CCHP system + PV unit + SEMCP-A2 |
| A3 | PV-gas-GSHP system: HESS + CCHP system+ PV unit + GSHPs + SEMCP-A3 |
| A4 | PV-gas-GSHP system: HESS + CCHP system + PV unit + GSHPs + SEMCP-A4 |

### 4.2. Data Acquisition

According to the method proposed in this paper, four parts of independent data were needed throughout the evaluation process. The first part was the value of quantitative criteria. The second part were the linguistic assessments on qualitative criteria made by decision makers. The other two parts were used to calculate the weights of evaluation criteria. The evaluation criteria system established in this paper included both quantitative criteria and qualitative criteria. The quantitative criteria referring to economy, energy and environment consisted of six sub-criteria that were described in detail in Section 3. The values of six quantitative sub-criteria associated to the four alternatives were collected directly from the offered bidding documents. All values of quantitative criteria were expressed in interval numbers, shown in Table 5. As for collecting the data of qualitative criteria, four decision makers were invited to make individual assessments of the four alternatives with respect to the six qualitative sub-criterion using the linguistic terms listed in Table 3. It is worthy to mention that all decision makers invited were experts in the IES field and the assessment results of linguistic variables for qualitative criteria are displayed in Table 6. Determining the weights of evaluation criteria was also an essential part in solving the MCDM problem. The neutrosophic entropy method was adopted in this paper to determine the weights of quantitative and qualitative criteria respectively. Decision

makers rated the importance of each evaluation criterion using the linguistic terms listed in Table 2. The linguistic rating results for quantitative sub-criteria weight are shown in Table 7, and those for qualitative sub-criteria weight are shown in Table 8. So all needed data were collected.

**Table 5.** The collected values for quantitative criteria.

|      | C1           | C2      | C3      | C4      | C5      | C6        |
| ---- | ------------ | ------- | ------- | ------- | ------- | --------- |
| A1   | [1050,1150]  | [40,43] | [36,41] | [22,24] | [45,48] | [4.8,5.2] |
| A2   | [1200,1350]  | [50,55] | [42,46] | [20,23] | [47,52] | [4.5,5]   |
| A3   | [1000,1100]  | [47,52] | [40,45] | [19,22] | [50,54] | [5.2,6]   |
| A4   | [1100,1200]  | [45,49] | [39,43] | [20,22] | [40,45] | [4.2,4.5] |

**Table 6.** Linguistic assessments for alternatives with respect to qualitative criteria made by decision makers.

|      |      | C7  | C8  | C9  | C10 | C11 | C12 |
| ---- | ---- | --- | --- | --- | --- | --- | --- |
| A1   | DM1  | G   | MG  | VG  | M   | G   | M   |
|      | DM2  | G   | M   | G   | VG  | G   | G   |
|      | DM3  | VG  | MG  | G   | G   | VG  | G   |
|      | DM4  | MG  | VG  | G   | M   | M   | G   |
| A2   | DM1  | G   | M   | MG  | G   | VG  | M   |
|      | DM2  | VG  | MG  | G   | MG  | M   | M   |
|      | DM3  | G   | VG  | G   | MG  | M   | G   |
|      | DM4  | G   | M   | M   | G   | G   | M   |
| A3   | DM1  | G   | M   | VG  | G   | VG  | M   |
|      | DM2  | VG  | G   | M   | G   | G   | MG  |
|      | DM3  | VG  | G   | G   | G   | VG  | MG  |
|      | DM4  | G   | VG  | G   | M   | MG  | VG  |
| A4   | DM1  | VG  | G   | M   | G   | G   | M   |
|      | DM2  | M   | MG  | G   | G   | M   | G   |
|      | DM3  | G   | M   | G   | G   | M   | VG  |
|      | DM4  | M   | G   | MG  | G   | G   | VG  |

**Table 7.** Linguistic ratings for quantitative sub-criteria weight made by decision makers.

|      | C1  | C2  | C3  | C4  | C5  | C6  |
| ---- | --- | --- | --- | --- | --- | --- |
| DM1  | VI  | VI  | I   | I   | M   | M   |
| DM2  | I   | VI  | I   | I   | M   | M   |
| DM3  | M   | I   | M   | M   | I   | M   |
| DM4  | M   | VI  | M   | I   | VI  | I   |

**Table 8.** Linguistic ratings for qualitative sub-criteria weight made by decision makers.

|      | C7  | C8  | C9  | C10 | C11 | C12 |
| ---- | --- | --- | --- | --- | --- | --- |
| DM1  | VI  | VI  | I   | I   | VI  | VI  |
| DM2  | I   | M   | I   | I   | I   | I   |
| DM3  | I   | I   | UI  | M   | I   | VI  |
| DM4  | M   | VI  | M   | I   | VI  | VI  |

*4.3. The Evaluation and Selection Process of the IES Construction Scheme Using the Extended TOPSIS Method*

4.3.1. Stage I: Calculation and Analysis of Quantitative Criteria

Step 1: Based on the data collected in Table 5, we constructed the quantitative criteria decision matrix and then normalized it. Among the six quantitative sub-criteria, C1

(construction cost) and C2 (operations and maintenance cost) are cost-type criteria that were normalized through Equation (7). C3 (primary energy conservation), C4 (renewable energy utilization), C5 ($CO_2$ emission reduction) and C6 (emission reduction of other pollutants) are benefit-type criteria that were normalized through Equation (8). The normalized quantitative decision matrix was obtained as follows.

$$
D_1^N = \begin{matrix} & C_1 & C_2 & C_3 & C_4 & C_5 & C_6 \\ A_1 \\ A_2 \\ A_3 \\ A_4 \end{matrix}
\begin{pmatrix}
[0.470, 0.567] & [0.524, 0.614] & [0.411, 0.522] & [0.483, 0.592] & [0.451, 0.526] & [0.461, 0.554] \\
[0.400, 0.496] & [0.409, 0.491] & [0.480, 0.585] & [0.439, 0.567] & [0.471, 0.569] & [0.432, 0.533] \\
[0.491, 0.595] & [0.433, 0.523] & [0.457, 0.572] & [0.417, 0.542] & [0.501, 0.591] & [0.500, 0.640] \\
[0.450, 0.541] & [0.460, 0.546] & [0.445, 0.547] & [0.439, 0.542] & [0.401, 0.493] & [0.404, 0.480]
\end{pmatrix}
$$

Step 2: Based on the normalized quantitative decision matrix, the relative positive ideal solution and the relative negative ideal solution for quantitative criteria were easily obtained by using Equations (9) and (10).

$y^+ = (y_1^+, y_2^+, y_3^+, y_4^+, y_5^+, y_6^+)$
$([0.491, 0.595], [0.524, 0.614], [0.480, 0.585], [0.483, 0.592], [0.501, 0.592], [0.500, 0.640])$
$y^- = (y_1^-, y_2^-, y_3^-, y_4^-, y_5^-, y_6^-)$
$([0.400, 0.496], [0.409, 0.491], [0.411, 0.522], [0.417, 0.542], [0.401, 0.492], [0.403, 0.480])$.

Step 3: According to the linguistic terms set along with the SVNNs in Table 2, the linguistic ratings for quantitative sub-criteria in Table 7 were quantified into the SVNNs shown in Table 9. Then, the neutrosophic entropy method was applied to calculate the criteria weights.

**Table 9.** SVNNs for quantitative sub-criteria weight.

|  | C1 | C2 | C3 | C4 | C5 | C6 |
|---|---|---|---|---|---|---|
| DM1 | <0.90,0.10,0.10> | <0.90,0.10,0.10> | <0.80,0.20,0.15> | <0.80,0.20,0.15> | <0.50,0.40,0.45> | <0.50,0.40,0.45> |
| DM2 | <0.80,0.20,0.15> | <0.90,0.10,0.10> | <0.80,0.20,0.15> | <0.80,0.20,0.15> | <0.50,0.40,0.45> | <0.50,0.40,0.45> |
| DM3 | <0.50,0.40,0.45> | <0.80,0.20,0.15> | <0.50,0.40,0.45> | <0.50,0.40,0.45> | <0.80,0.20,0.15> | <0.50,0.40,0.45> |
| DM4 | <0.50,0.40,0.45> | <0.90,0.10,0.10> | <0.50,0.40,0.45> | <0.80,0.20,0.15> | <0.90,0.10,0.10> | <0.80,0.20,0.15> |

By applying Equation (4), we have:

$$
E(C_j) = 1 - \frac{1}{4} \sum_{i=1}^{4} \left\{ (T_{C_j}(x_i) + F_{C_j}(x_i)) \cdot \left| I_{C_j}(x_i) - I_{C_j^c}(x_i) \right| \right\}, j = 1, 2, \cdots, 6.
$$

Therefore, the neutrosophic entropy of six quantitative sub-criteria were computed, $E(C_1) = 0.5625, E(C_2) = 0.2575, E(C_3) = 0.62, E(C_4) = 0.525, E(C_5) = 0.5625, E(C_6) = 0.715$.

According to Equation (5), the weights of six quantitative sub-criteria were obtained as $\omega' = (\omega_1, \omega_2, \omega_3, \omega_4, \omega_5, \omega_6) = (0.159, 0.269, 0.138, 0.172, 0.159, 0.103)$.

Step 4: By utilizing Equations (11) and (12), the weighted normalized Euclidean distance measure of four alternatives from the relative positive ideal solution (RPIS) for quantitative criteria $d^{I+}$ and that from the relative negative ideal solution (RNIS) $d^{I-}$ were calculated. Then, the relative closeness coefficient $c^I$ to the ideal solution for quantitative criteria was also calculated through Equation (13). Calculation results for quantitative criteria are shown in Table 10.

**Table 10.** The calculation result of, $d^{I+}$, $d^{I-}$ and $c^I$.

| Alternative | $d^{I+}$ | $d^{I-}$ | $c^I$ |
|---|---|---|---|
| A1 | 0.016814 | 0.031354 | 0.65093 |
| A2 | 0.032593 | 0.017044 | 0.343373 |
| A3 | 0.021843 | 0.029812 | 0.577137 |
| A4 | 0.030192 | 0.014526 | 0.324836 |

4.3.2. Stage II: Calculation and Analysis for Qualitative Criteria

Step 1: In order to obtain the qualitative criteria decision matrix, the linguistic data collected in Table 6 were quantified into SVNNs according to the linguistic term set along with SVNNs in Table 3. Table 11 shows assessment results of SVNNs transferred from the linguistic variables in Table 6.

**Table 11.** SVNNs of linguistic assessment for qualitative criteria.

|  |  | C7 | C8 | C9 | C10 | C11 | C12 |
|---|---|---|---|---|---|---|---|
| A1 | DM1 | <0.80,0.20,0.15> | <0.65,0.35,0.30> | <0.90,0.10,0.05> | <0.50,0.50,0.45> | <0.80,0.20,0.15> | <0.50,0.50,0.45> |
|  | DM2 | <0.80,0.20,0.15> | <0.50,0.50,0.45> | <0.80,0.20,0.15> | <0.90,0.10,0.05> | <0.80,0.20,0.15> | <0.80,0.20,0.15> |
|  | DM3 | <0.90,0.10,0.05> | <0.65,0.35,0.30> | <0.80,0.20,0.15> | <0.80,0.20,0.15> | <0.90,0.10,0.05> | <0.80,0.20,0.15> |
|  | DM4 | <0.65,0.35,0.30> | <0.90,0.10,0.05> | <0.80,0.20,0.15> | <0.50,0.50,0.45> | <0.50,0.50,0.45> | <0.80,0.20,0.15> |
| A2 | DM1 | <0.80,0.20,0.15> | <0.50,0.50,0.45> | <0.65,0.35,0.30> | <0.80,0.20,0.15> | <0.90,0.10,0.05> | <0.50,0.50,0.45> |
|  | DM2 | <0.90,0.10,0.05> | <0.65,0.35,0.30> | <0.80,0.20,0.15> | <0.65,0.35,0.30> | <0.50,0.50,0.45> | <0.50,0.50,0.45> |
|  | DM3 | <0.80,0.20,0.15> | <0.90,0.10,0.05> | <0.80,0.20,0.15> | <0.65,0.35,0.30> | <0.50,0.50,0.45> | <0.80,0.20,0.15> |
|  | DM4 | <0.80,0.20,0.15> | <0.50,0.50,0.45> | <0.50,0.50,0.45> | <0.80,0.20,0.15> | <0.80,0.20,0.15> | <0.50,0.50,0.45> |
| A3 | DM1 | <0.80,0.20,0.15> | <0.50,0.50,0.45> | <0.90,0.10,0.05> | <0.80,0.20,0.15> | <0.90,0.10,0.05> | <0.50,0.50,0.45> |
|  | DM2 | <0.90,0.10,0.05> | <0.80,0.20,0.15> | <0.50,0.50,0.45> | <0.80,0.20,0.15> | <0.80,0.20,0.15> | <0.65,0.35,0.30> |
|  | DM3 | <0.90,0.10,0.05> | <0.80,0.20,0.15> | <0.80,0.20,0.15> | <0.80,0.20,0.15> | <0.90,0.10,0.05> | <0.65,0.35,0.30> |
|  | DM4 | <0.80,0.20,0.15> | <0.90,0.10,0.05> | <0.80,0.20,0.15> | <0.50,0.50,0.45> | <0.65,0.35,0.30> | <0.90,0.10,0.05> |
| A4 | DM1 | <0.90,0.10,0.05> | <0.80,0.20,0.15> | <0.50,0.50,0.45> | <0.80,0.20,0.15> | <0.80,0.20,0.15> | <0.50,0.50,0.45> |
|  | DM2 | <0.50,0.50,0.45> | <0.65,0.35,0.30> | <0.80,0.20,0.15> | <0.80,0.20,0.15> | <0.50,0.50,0.45> | <0.80,0.20,0.15> |
|  | DM3 | <0.80,0.20,0.15> | <0.50,0.50,0.45> | <0.80,0.20,0.15> | <0.80,0.20,0.15> | <0.50,0.50,0.45> | <0.90,0.10,0.05> |
|  | DM4 | <0.50,0.50,0.45> | <0.80,0.20,0.15> | <0.65,0.35,0.30> | <0.80,0.20,0.15> | <0.80,0.20,0.15> | <0.90,0.10,0.05> |

Step 2: For determining the weight of decision makers, the importance of decision makers denoted by linguistic term along with SVNNs is collected in Table 12. By utilizing Equation (14), the weight of the first decision makers was computed as follows:

$$e_1 = \frac{1 - \sqrt{\left\{ (1-0.9)^2 + (0.1) + (0.1) \right\}/3}}{\sum\limits_{k=1}^{4} \left\{ 1 - \sqrt{\left\{ (1-0.9)^2 + (0.1) + (0.1) \right\}/3} \right\}} = 0.285.$$

**Table 12.** Importance assessment of decision makers expressed with SVNNs.

| Decision Maker | DM1 | DM2 | DM3 | DM4 |
|---|---|---|---|---|
| Linguistic term | VI | M | VI | I |
| SVNN | <0.90,0.10,0.10> | <0.50,0.40,0.45> | <0.90,0.10,0.10> | <0.80,0.20,0.15> |

Similarly, the weights of the other three decision makers were obtained,

Step 3: In this step, by using Equation (15), we aggregated the qualitative criteria decision matrix with decision makers' weight obtained. A part of the calculation procedure shows as follows:

$$d_{11} = \langle T_{11}, I_{11}, F_{11} \rangle$$
$$T_{11} = 1 - (1 - 0.8)^{0.285} \times (1 - 0.8)^{0.173} \times (1 - 0.9)^{0.285} \times (1 - 0.65)^{0.257} = 0.810$$
$$I_{11} = (0.2)^{0.285} \times (0.2)^{0.173} \times (0.1)^{0.285} \times (0.35)^{0.257} = 0.190$$
$$F_{11} = (0.15)^{0.285} \times (0.15)^{0.173} \times (0.05)^{0.285} \times (0.3)^{0.257} = 0.131$$

Then, neutrosophic decision matrix that aggregates decision makers' weight is shown in Table 13.

**Table 13.** The aggregated SVNNs-based decision matrix.

| | C7 | C8 | C9 | C10 | C11 | C12 |
|---|---|---|---|---|---|---|
| A1 | <0.810,0.190,0.131> | <0.730,0.270,0.203> | <0.836,0.164,0.110> | <0.709,0.291,0.225> | <0.792,0.208,0.145> | <0.740,0.260,0.205> |
| A2 | <0.822,0.177,0.124> | <0.703,0.297,0.224> | <0.703,0.297,0.242> | <0.742,0.258,0.206> | <0.750,0.250,0.181> | <0.615,0.385,0.329> |
| A3 | <0.854,0.146,0.091> | <0.783,0.217,0.155> | <0.808,0.192,0.133> | <0.747,0.253,0.199> | <0.844,0.156,0.096> | <0.719,0.281,0.212> |
| A4 | <0.757,0.243,0.176> | <0.713,0.286,0.231> | <0.700,0.300,0.245> | <0.800,0.200,0.150> | <0.696,0.304,0.248> | <0.821,0.178,0.113> |

Step 4: According to Equations (16) and (17), the relative positive ideal solution and the relative negative ideal solution $y^{N-}$ for qualitative criteria were easily obtained as follows:

$$y^{N+} = \{y_6^+, y_7^+, \cdots, y_{12}^+\} = \{\langle 0.854, 0.146, 0.091 \rangle, \langle 0.783, 0.217, 0.155 \rangle, \langle 0.836, 0.164, 0.110 \rangle,$$
$$\langle 0.8, 0.2, 0.15 \rangle, \langle 0.844, 0.156, 0.096 \rangle, \langle 0.822, 0.178, 0.113 \rangle\}$$
$$y^{N-} = \{y_6^-, y_7^-, \cdots, y_{12}^-\} = \{\langle 0.757, 0.243, 0.176 \rangle, \langle 0.703, 0.297, 0.231 \rangle, \langle 0.700, 0.300, 0.245 \rangle,$$
$$\langle 0.709, 0.291, 0.225 \rangle, \langle 0.696, 0.304, 0.248 \rangle, \langle 0.615, 0.385, 0.329 \rangle\}.$$

Step 5: Based on the data collected in Table 8, the weights of qualitative criteria were determined the same as the procedure for quantitative criteria. Then, the weight of six qualitative sub-criteria was acquired as follows:

$$\omega'' = (\omega_7, \omega_8, \omega_9, \omega_{10}, \omega_{11}, \omega_{12}) = (0.156, 0.173, 0.113, 0.139, 0.201, 0.218)$$

Step 6: By utilizing Equations (18) and (19), the weighted normalized Euclidean distance measure of four alternatives from the relative positive ideal solution (RPIS) for qualitative criteria $d^{N+}$ and that from the relative negative ideal solution (RNIS) $d^{N-}$ were calculated. Then, the relative closeness coefficient $c^N$ to the ideal solution for qualitative criteria was also calculated through Equation (20). All calculation results for qualitative criteria are shown in Table 14.

**Table 14.** The calculation result of $d^{N+}$, $d^{N-}$ and $c^N$.

| Alternative | $d^{N+}$ | $d^{N-}$ | $c^N$ |
|---|---|---|---|
| A1 | 0.025433 | 0.036453 | 0.58903 |
| A2 | 0.049846 | 0.015338 | 0.235307 |
| A3 | 0.021193 | 0.042908 | 0.669384 |
| A4 | 0.038411 | 0.04213 | 0.523087 |

### 4.3.3. Stage III: Acquisition of the Comprehensive Relative Closeness Coefficient and Ranking the Alternatives

Based on the relative closeness coefficient $c^I$ for quantitative criteria in Table 10 and the relative closeness coefficient $c^N$ for qualitative criteria in Table 14, the comprehensive relative closeness coefficient $c$ was obtained by using Equation (21) when the fusion coefficient $\mu$ equaled 0.5. Then, the comprehensive relative closeness coefficient $c$ and the

ranking results of the alternatives are shown in Table 15. As we can see, A3 was the best construction scheme among the four alternatives.

**Table 15.** The comprehensive relative closeness coefficient $c$ and ranking results ($\mu$ = 0.5).

| Alternative | $c^I$ | $c^N$ | $c$ | Ranking |
|---|---|---|---|---|
| A1 | 0.65093 | 0.58903 | 0.61998 | 2 |
| A2 | 0.343373 | 0.235307 | 0.28934 | 4 |
| A3 | 0.577137 | 0.669384 | 0.62326 | 1 |
| A4 | 0.324836 | 0.523087 | 0.423962 | 3 |

## 5. Discussion

### 5.1. Scenario Analysis

Considering the improvement of technological capabilities and service level in recent years, technology and service criteria as innovative qualitative criteria were added to the evaluation system of the IES performance. The evaluation criteria system was then divided into two parts: quantitative criteria and qualitative criteria. Each part was analyzed and calculated respectively. Then, as Equation (21) showed, the comprehensive relative closeness coefficient $c_i$ was obtained through fusing the quantitative criteria relative closeness coefficient $c_i^I$ and the qualitative criteria relative closeness coefficient $c_i^N$ with the fusion coefficient $\mu$. Obviously, the value of $\mu$ had an influence on the evaluation result. Specifically, the greater $\mu$ was, more weight assigned to the traditional quantitative criteria. Furthermore, when the value of $\mu$ was more than 0.5, the traditional quantitative criteria was considered more important than the innovative qualitative criteria in the evaluation system. However, when the value of $\mu$ was less than 0.5, the innovative qualitative criteria was considered more important. Consequently, the extended TOPSIS method was able to balance the importance of traditional quantitative criteria against the innovative qualitative criteria by adjusting the value of the fusion coefficient $\mu$.

The evaluation criteria system has a significant effect on leading the application and development of the IES. Taking into account the imbalance of regional development, the importance of traditional quantitative criteria and innovative qualitative criteria should be carefully determined to accommodate the local IES development stages. In this section, we discuss two different application scenarios: the practical-oriented scenario and the innovation-oriented scenario.

Scenario I (the practical-oriented scenario): When the area is at the early stage of developing the IES, the energy supplier pays more attention to the practicality and economic efficiency of the IES construction scheme. Therefore, an economical, environmentally friendly and energy-saving IES construction scheme is most wanted. Then, the decision maker should relatively increase the weight of traditional quantitative criteria (economy, energy and environment criteria) in the evaluation criteria system. So the decision makers assign to the fusion coefficient $\mu$ a value greater than 0.5.

As we can see from Figure 1, when the value of the fusion coefficient $\mu$ is 0.5, the ranking result of the alternatives is $A3 > A1 > A4 > A2$ and A3 scores slightly better than A1. However, as the value of $\mu$ gradually increases, the score of alternative A1 surpasses A3. In other words, when $\mu$ is assigned a value over 0.6, alternative A1 is ranked first and selected to be the most appropriate construction scheme. As the value of $\mu$ increases, the alternative A4 curve shows a downward trend, which means its performance on traditional quantitative criteria is worse than its performance on innovative qualitative criteria. In contrast, alternative A2 curve shows an upward trend. But the overall performance of A2 always ranks the worst.

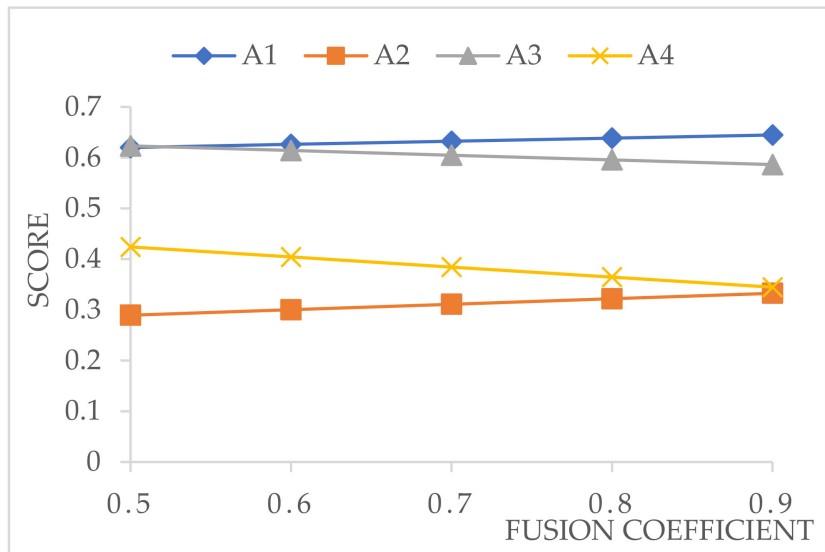

**Figure 1.** Scenario I analysis: the practical-oriented scenario.

Scenario II (innovation-oriented scenario): When the area is at a mature stage of the IES application, the energy supplier would like to give more attention to the innovative qualitative evaluation criteria. Therefore, we should relatively increase the importance of qualitative evaluation criteria in the evaluation criteria system. At this time, the requirements for the IES construction scheme are not merely to meet the economic, environmental and energy demand, but also to better technological innovation and service quality improvement to users. Then, the decision makers set the value of the fusion coefficient $\mu$ to be less than 0.5.

Similarly, it can be seen from Figure 2 that when the fusion coefficient is 0.5, the overall performance of A3 is slightly better than A1. As the fusion coefficient gradually decreases, the score of A1 decreases, while the score of alternative A3 increases instead, making the gap expand. It means that alternative A3 has better performance on innovative qualitative criteria, while A1 performs relatively poorly on qualitative criteria. On this condition, the A4 upward curve means that it has better performance on qualitative criteria than quantitative criteria, which is consistent with the conclusion obtained in Scenario I.

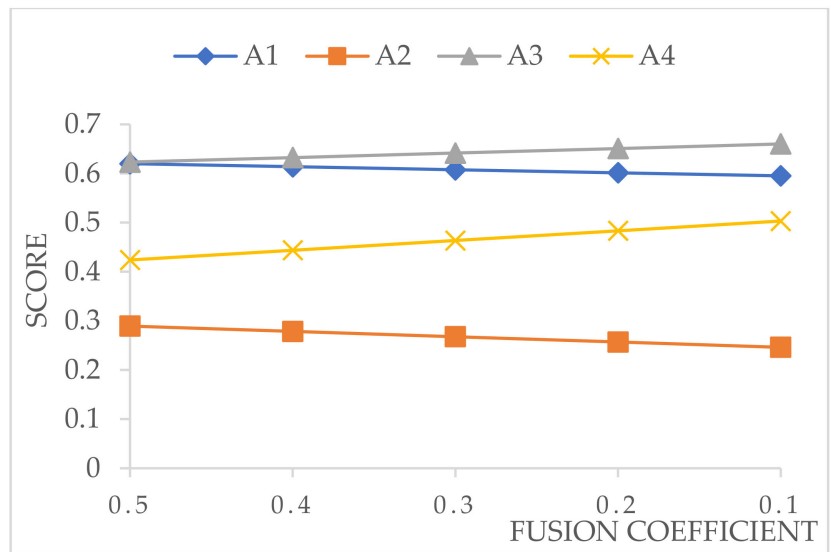

**Figure 2.** Scenario II analysis: the innovation-oriented scenario.

*5.2. Sensitivity Analysis*

In order to verify the stability of the extended TOPSIS method, sensitivity analysis on the weight of each sub-criterion was conducted. In order to maintain the independence between quantitative criteria and qualitative criteria, when we performed the sensitivity analysis on the weight of a quantitative sub-criterion, the weights of qualitative criteria remained unchanged, and vice versa.

As shown in Figures 3 and 4, we reduced the weight of a sub-criterion by 10% and 20% or increased it by 10% and 20%, compared to the base weight, for sensitivity analysis. The result shows that alternative A3 had a higher score than alternative A1 in most cases, no matter how the weight was adjusted. However, there were a few cases where the score of alternative A1 slightly exceeded alternative A3 when the weight of a sub-criterion was adjusted in a specific direction. The main reason is that the overall scores of alternative A3 and alternative A1 were very close when all criteria were on the base weight. As a result, when the weight of a certain sub-criterion positive to A1 increased to a certain extent, the overall score of A1 might surpass A3. As we can see, the score curve of the alternatives is almost parallel to the coordinate axis throughout sensitivity analysis. This shows that the weight of a single sub-criterion fluctuated within a specific range and had little influence on the evaluation results. In conclusion, it verified that the evaluation method has good stability and robustness.

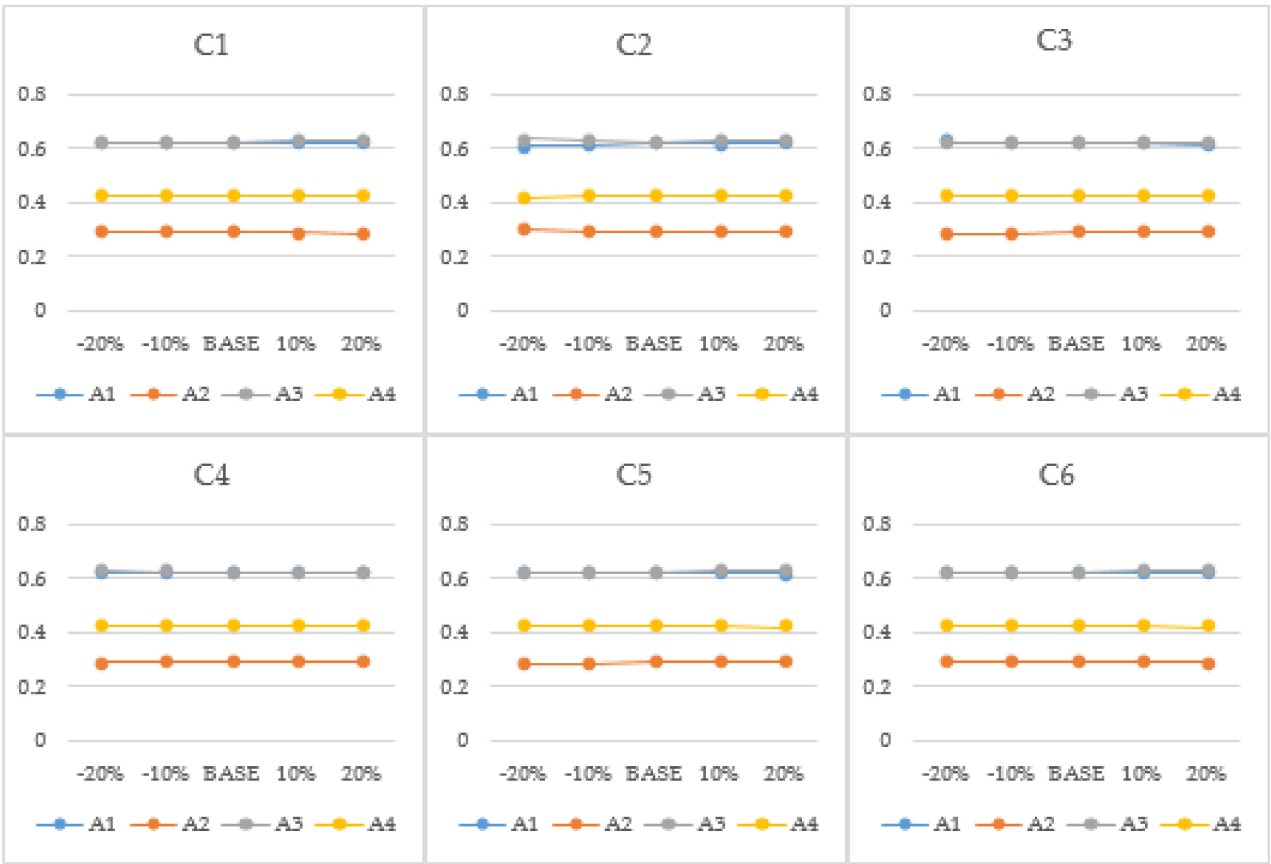

**Figure 3.** Sensitivity analysis for quantitative criteria.

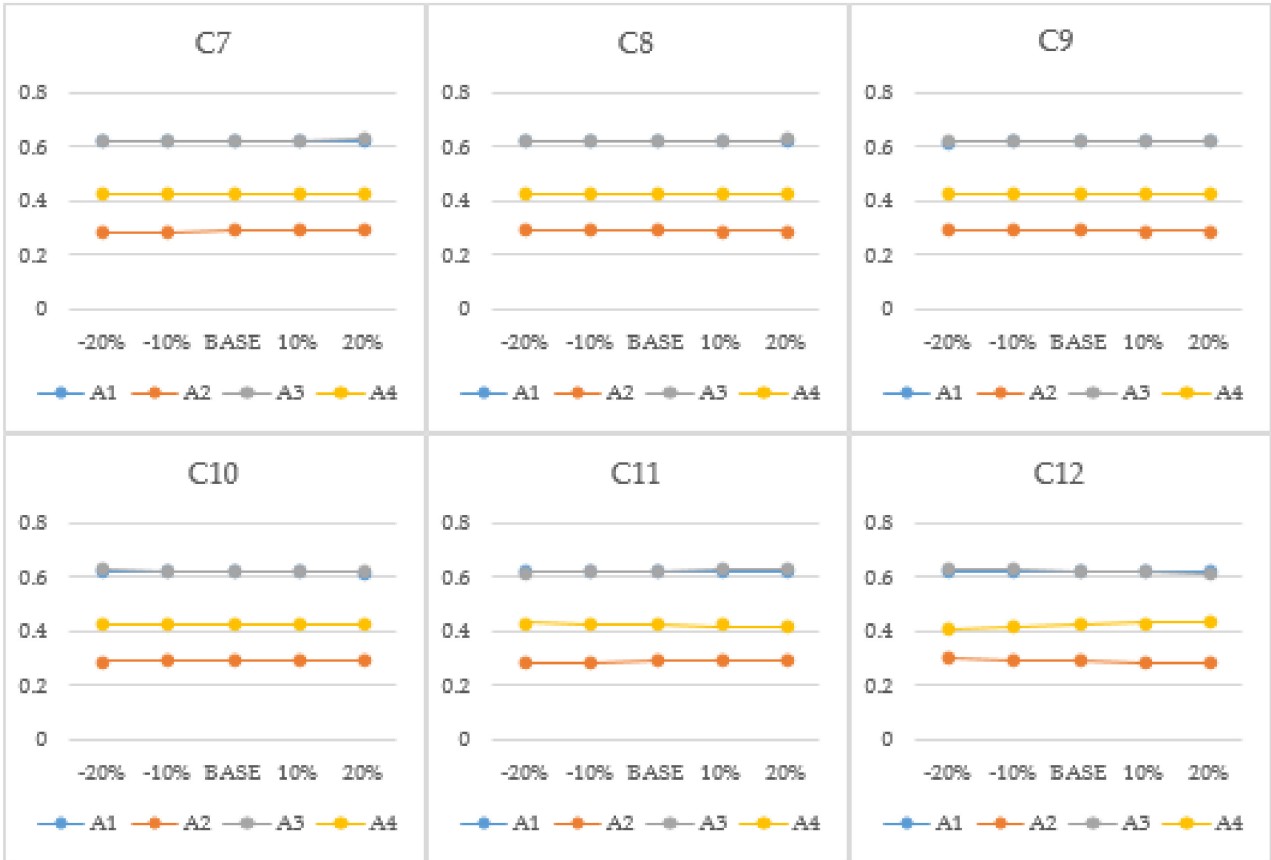

**Figure 4.** Sensitivity analysis for qualitative criteria.

### 5.3. Comparative Analysis

The VIKOR method is also widely adopted to solve multi-criteria decision-making problems. Different from the TOPSIS method, which determines a solution with the shortest distance from the ideal solution and the farthest distance from the negative ideal solution, the VIKOR method of compromise ranking determines a compromise solution, providing a maximum "group utility" for the "majority" and a minimum of an individual regret for the "opponent".

In order to verify the reliability of the evaluation result, an extended VIKOR method proposed by Liu [49] was applied to evaluate the same case in this paper. The evaluation result is shown in Table 16.

**Table 16.** Evaluation results by an extended VIKOR method.

|  | $S_i$ | $R_i$ | $Q_i$ | Ranking |
|---|---|---|---|---|
| A1 | 0.47585 | 0.0559 | 0.03624 | 2 |
| A2 | 0.72654 | 0.0865 | 1 | 4 |
| A3 | 0.45626 | 0.0566 | 0.011438 | 1 |
| A4 | 0.64559 | 0.0758 | 0.675411 | 3 |

It can be seen from that the ranking of the evaluation the result obtained by the extended VIKOR was $A3 > A1 > A4 > A2$, which was consistent with the result obtained by the proposed method in this paper. Therefore, the reliability of the evaluation result of the proposed method in this paper is verified.

## 6. Conclusions

Considering the communication and information technology, big data technology, Internet of Things technology and artificial intelligence technology that apply to the SEMCP embedded in the IES, this paper constructed a comprehensive criteria system for evaluating the park-level IES equipped with the SEMCP. Besides the traditional quantitative evaluation criteria, technology criteria and service criteria as innovative qualitative criteria were added to evaluate the SEMCP in the IES. SVNN was introduced to evaluate the IES construction scheme for the first time. In this paper, SVNN and its related method were used three times in total in the entire evaluation process. It showed SVNN has a flexible usage in the MCDM method. An extended TOPSIS method was proposed to accommodate the mixed data types of the evaluation criteria system. Notably, using the fusion coefficient $\mu$ to integrate two respective calculation results of quantitative and qualitative criteria is one key step of the extended TOPSIS method. Decision makers can adjust the value of $\mu$ to satisfy the requirements of the energy supplier. By applying the proposed method to solving a real case in China, the evaluation procedure of the IES construction scheme proceeded well and the ranking result was $A3 > A1 > A4 > A2$, where $\mu = 0.5$. The ranking result showed alternative A3 to be the most appropriate IES construction scheme for the case in this paper. Then, a comprehensive analysis was conducted according to the evaluation result. For scenario analysis, we analyzed different application scenarios from the view of the IES development stage. For sensitivity analysis, whenever the weight of the criteria fluctuated in a small range independently, the ranking result stayed basically unchanged. It verified the stability and robustness of the evaluation result. For comparative analysis, an extended VIKOR method was applied to solve the problem in this paper. The ranking result was consistent with the result obtained by the extended TOPSIS method proposed. Then, the reliability of the proposed method is verified. In short, this study provides a useful tool for the energy supplier to evaluate and select a preferred IES construction scheme.

The idea of dividing the evaluation criteria system into qualitative criteria and quantitative criteria to analyze separately can also be applied to MCDM problems in other fields. But sometimes the qualitative criteria and quantitative criteria when divided may not be completely independent of each other. The issue of independence was not considered and analyzed when determining criteria weight in this paper. This needs to receive more attention in further research.

**Author Contributions:** Conceptualization, W.Z.; methodology, J.W.; validation, W.Z.; formal analysis, L.Q.; investigation, J.W.; resources, P.Y.; data curation, W.Z.; writing—original draft preparation, J.W.; writing—review and editing, W.Z.; visualization, L.Q.; supervision, J.W.; project administration, L.Q.; funding acquisition, P.Y. All authors have read and agreed to the published version of the manuscript.

**Funding:** This research was funded by The Ministry of Education of the Humanities and Social Science project, the National Natural Science Foundation of China and the Scientific Research Project of Liaoning Provincial Department of Education, grant numbers 18YJAZH138, 20BJL036 and LG201912. The APC was funded by The Ministry of Education of the Humanities and Social Science project, the National Natural Science Foundation of China and the Scientific Research Project of Liaoning Provincial Department of Education, grant numbers 18YJAZH138, 20BJL036 and LG201912.

**Institutional Review Board Statement:** Not applicable.

**Informed Consent Statement:** Not applicable.

**Data Availability Statement:** Not applicable.

**Conflicts of Interest:** The authors declare no conflict of interest.

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
