# Peer review of "Evaluation and Selection of Integrated Energy System Construction Scheme Equipped with Smart Energy Management and Control Platform Using Single-Valued Neutrosophic Numbers"

_sustainability, doi:10.3390/su13052615_

Round 1
Reviewer 1 Report
The paper is a precious contribution to have a conceptual framework fighting against the climate change issue
Very intersting and multidisciplinary work. Some terms and concepts (e.g. IES, TOPSIS...) need more explanation to better understand the aim of the paper.
is well written and
Reviewer 2 Report
The authors propose “Evaluation and Selection of Integrated Energy System Construction Scheme Equipped with Smart Energy Management and Control Platform Using Single-valued Neutrosophic Numbers.” The topic presented in the paper is interesting. The following concerns are communicated with the authors for improvement.
1. Please define IES and TOPSIS in the abstract section.
2. Please remove too much jargon and acronyms from the abstract. The abstract section is very long; it should clearly state the main contribution of this study.
3. The introduction section is very long. I suggest dividing this into several subsections.
4. The second last paragraph summarizes the main contribution of this work. However, the main research questions are not defined before writing contributions.
5. In the introduction section, the authors need to discuss the existing literature critically to clearly show the research gap. Thus, first, find the gap and then write the contribution.
6. In section 2.3, criteria 5 (C5), the utilization of renewable energy will have a negative impact on system reliability since such a source is intermittent in nature. Thus, this criterion should be designed properly. Furthermore, the higher the penetration of renewable energy lower the power system inertia (see 10.1109/ACCESS.2020.3031481 and 10.1016/j.rser.2019.109369 for details.). It is true that renewable energy utilization reduces carbon emission; however, these technical issues cannot be ignored.
7. In Section 2.4, criteria 8 (C8), the economic dispatch (ED) is not possible without the unit commitment (UC). In UC, depending on the available units and load demand, the economically feasible units are brought online to meet the real-time demand. Thus, the authors need to carefully consider this fact.
8. Some references are not properly placed. For example, the reference [39]
9. Fuzzification in eq. 1 is either continuous or discrete? kindly clarify and incorporate corrections.
10. How the limits of the linguistic terms are defined in table 2? I would suggest putting these values in graphical format instead of tabular format. The same is true for other tables, e.g. table 3 and so on.
11. The problem statement should not be under the results section.
12. In section 5.3, a comparative study is performed. Please add more details about the VIKOR method. Also, what are the reasons that make the proposed method better than VIKOR method?
13. The conclusion section is very long. It must be reformulated to reduce the text. For example, the first paragraph of the conclusion section can be completely removed.
Overall, the contents presented in the paper is good. The authors should address the comments to improve the quality of the manuscript.
Round 2
Reviewer 2 Report
Thank you for addressing all comments carefully. The quality of the manuscript has been enhanced significantly. I suggest modifying the existing title "Evaluation and Selection of Integrated Energy System Con
struction Scheme Equipped with Smart Energy Management
and Control Platform Using Single-valued Neutrosophic Num
bers". In my opinion, the title should be concise and short to get a clear idea about the topic and contribution of the works.